# Self-assembly of polycyclic supramolecules using linear metal-organic ligands

Bo Song[1], Sneha Kandapal[2], Jiali Gu[3], Keren Zhang[2], Alex Reese[2], Yuanfang Ying[4], Lei Wang[1], Heng Wang [1], Yiming Li[1], Ming Wang[5], Shuai Lu[6], Xin-Qi Hao[6], Xiaohong Li [3], Bingqian Xu[2] & Xiaopeng Li[1]

Coordination-driven self-assembly as a bottom-up approach has witnessed a rapid growth in building giant structures in the past few decades. Challenges still remain, however, within the construction of giant architectures in terms of high efficiency and complexity from simple building blocks. Inspired by the features of DNA and protein, which both have specific sequences, we herein design a series of linear building blocks with specific sequences through the coordination between terpyridine ligands and Ru(II). Different generations of polycyclic supramolecules (**C1** to **C5**) with increasing complexity are obtained through the self-assembly with Cd(II), Fe(II) or Zn(II). The assembled structures are characterized via multi-dimensional mass spectrometry analysis as well as multi-dimensional and multinuclear NMR ($^1$H, COSY, NOESY) analysis. Moreover, the largest two cycles **C4** and **C5** hierarchically assemble into ordered nanoscale structures on a graphite based on their precisely-controlled shapes and sizes with high shape-persistence.

[1] Department of Chemistry, University of South Florida, Tampa, FL 33620, USA. [2] Single Molecule Study Laboratory, College of Engineering and Nanoscale Science and Engineering Center, University of Georgia, Athens, GA 30602, USA. [3] College of Chemistry, Chemical Engineering and Materials Science, Soochow University, Suzhou 215123, China. [4] Department of Chemistry and Biochemistry, Texas State University, San Marcos, TX 78666, USA. [5] State Key Laboratory of Supramolecular Structure and Materials, College of Chemistry, Jilin University, Changchun, Jilin 130012, China. [6] College of Chemistry and Molecular Engineering, Zhengzhou University, Zhengzhou, Henan 450001, China. These authors contributed equally: Bo Song, Sneha Kandapal. Correspondence and requests for materials should be addressed to X.-Q.H. (email: xqhao@zzu.edu.cn) or to X.L. (email: lxh83@suda.edu.cn) or to B.X. (email: bxu@engr.uga.edu) or to X.L. (email: xiaopengli1@usf.edu)

Self-assembly as one of the main organizing principles of biological systems is also a widely applied strategy in supramolecular chemistry as the driving forces for the assembly of artificial structures from simple building blocks[1–6]. Among the diverse fields of supramolecular chemistry, coordination-driven self-assembly offers a 'bottom-up' approach to mimic nature's activities and constructs various 2D and 3D metallo-supramolecules based on the highly directional and predictable feature of coordination[7–22]. Indeed, the design of metallo-supramolecules has matured beyond the proof of principles and is ready to face more challenges with respect to the complexity of assembled architectures rather than relatively simple polygons and polyhedrons.

In coordination-driven self-assembly, the design and self-assembly were heavily concentrated on geometrical parameters (e.g., the size[23], angularity[24, 25], and dimensionality of ligand) and influence of temperature[26], solvent[27], and counterions[28–32]. In biological systems, the shapes, complexity, and functions of DNA- and protein-based assemblies are encoded by the defined sequences of nucleotides and amino acids[33, 34]. The recent success of DNA nanostructure[35], e.g., DNA origami assembled by specific sequence of DNA chains[36, 37], further inspired us to revisit the coordination-driven self-assembly. We envisioned that linear building blocks with specific sequence could facilitate the self-assembly of metallo-supramolecules with increasing complexity.

Thanks to the diversity of coordination-driven self-assembly, we herein report a series of linear building blocks with specific sequence through bridging terpyridine ligands using Ru(II) coordination with high stability. In the following self-assembly, Zn(II), Fe(II), and Cd(II) metal ions with weak coordination but high reversibility are used to assemble discrete 2D fractal architectures[38] ranging from generation 1 (**C1**) to generation 5 (**C5**), from 3,360 Da to 38,066 Da with precisely-controlled structures and increasing complexity (Fig. 1). Moreover, such metallo-supramolecules with precisely-controlled shapes and sizes can serve as promising building blocks to further assemble into ordered nanoscale structures on a solid surface, given that surface self-assembly has attracted considerable attention in materials science[39–43].

## Results

### Three approaches in the design and synthesis of ligands L1–L5.

A series of linear metal-organic building blocks was first designed through the connections of two basic residues **A** and **B** with Ru (II) (Fig. 2). In this system, '**A**' provided an expansion of the structure, while '**B**' served as end-cap which led to the formation of discrete architectures. For example, the terminator '**B**' could be solely used as a ligand to construct the smallest structure **C1**; while **L4**, which has the sequence '**AB**' would self-assemble to **C4** with weak coordination metal ions. Meanwhile, elongation of the '**AB**' sequence to either side gave different ligands, and thus changed the shape and complexity of the final supramolecules. The sequence '**ABB**' would assemble **C3** with *C3* symmetry, but the '**BBABB**' sequence, which was created via elongation with **B** on both sides, would only construct a smaller structure **C2** with *C2* symmetry. On the contrary, by introducing '**A**' on the left and '**B**' on the right to form the '**AABB**' sequence, the ligand for a high generation of supramolecule **C5** could be created. However, it should be noted that the actual synthesis of ligands was not approached by linking '**A**' and '**B**' with Ru(II) directly. Basically, the preparation of terpyridine-Ru(II) metal-organic ligands could be categorized into three approaches as shown in Supplementary Figure 1, including (1) end-capping approach based on the coordination with Ru(III) complex followed by reduction; (2) Suzuki coupling reaction on terpyridine-Ru(II) complex; (3) Sonogashira coupling reaction on terpyridine-Ru(II) complex. End-capping approach was first developed to prepare symmetric metal-organic ligand. With the introduction of Suzuki coupling reaction, short linear metal-organic ligand could be obtained for single point or multiple points with the same -B(OH)$_2$ reactant. Without appropriate protection and deprotection groups, however, it is very challenging to use Suzuki coupling reaction for longer asymmetric metal-organic ligand synthesis. Subsequently, Sonogashira coupling reaction on the terpyridine-Ru(II) complexes, which include–TMS protection and deprotection was used to define the linear sequence of metal-organic building blocks with rigid linkages (Supplementary Figures 3–7). This synthetic approach is reminiscent of the protection and deprotection in peptide synthesis.

### Synthesis of ligands L1-L5, self-assembly of supramolecules C1-C5, and characterization with NMR and mass spectrometry.

Our study was initiated from the construction of **C1**. Alkyl chains were used as substituent groups to improve solubility. Not surprisingly, treatment of pure organic ligand **L1** with Zn(II) in 1:1 ratio led to the formation of multiple species (Supplementary Figure 14), due to the flexibility of single-layered ligands[44]. Following the same route of previous study[19], Fe(II) was used instead, which has a stronger binding with terpyridine[45]. After column chromatography, **C1** was obtained in 39% yield, and the structure was further characterized by electrospray ionization–mass spectrometry (ESI-MS) and traveling wave ionmobility–mass spectrometry (TWIM-MS)[44] (Supplementary Figure 15), as well as multi-dimensional NMR (Supplementary Figures 63–68).

Next, **L4** with sequence '**AB**' was synthesized to construct **C4**. As has been depicted before, a stepwise strategy of coordinating Ru(III) complex with '**A**' first, followed by Sonogashira coupling with alkyne-terpyridine was applied to give **L4**. Then **C4** was assembled by mixing **L4** and Zn(II) in 1:2 ratio in CHCl$_3$/MeOH for 8 h. For the zinc-coordinated terpyridines, all 3',5' protons showed downfield shift on $^1$H NMR, while the 6,6" protons shifted upfield (Supplementary Figure 109). The aliphatic region displayed two sets of peaks from 0.2 ppm to 4.0 ppm, corresponding to the alkyl chain on the inner and outer of the ring (Supplementary Figure 90). $^{19}$F NMR showed two sets of peaks at −114.64 ppm, −114.81 ppm, respectively, showing a highly symmetric assembly (Supplementary Figure 92). Meanwhile, ESI-MS and TWIM-MS were both used for characterization. A series of peaks with continuous charge states ranging from 8+ to 21+ were recorded on ESI-MS (Fig. 3c). Each charge state showed well-resolved isotope patterns, which were consistent with theoretical isotope distribution (Supplementary Figure 21). After deconvolution, the molecular weight was 18470 Da, in well agreement with the chemical composition of C$_{780}$H$_{576}$F$_{120}$P$_{18}$N$_{108}$O$_{12}$Ru$_6$Zn$_{12}$. TWIM-MS showed one set of peaks from 12+ to 21+ with narrow drift time distribution, excluding the formation of other isomeric structures (Fig. 3d).

The journey proceeded to **C3** with the construction of '**ABB**' sequence. Being similar to **L4**, **L3** was also synthesized by the direct coupling reaction between two Ru complexes (Supplementary Figure 6). Note that the inner part of **L3** was similar to that of **L1** in terms of size and angle. However, after mixing **L3** and Zn (II) in 1:2 ratio at 50 °C for 8 h, a single species with only one set of peaks from 5+ to 12+ was observed on ESI-MS, and the molecular weight after deconvolution was consistent with [**L3**$_3$Zn$_6$(PF$_6$)$_{12}$] (Fig. 3e). TWIM-MS also showed only one series of peaks (Fig. 3f), suggesting the success formation of the desired product with rigid structure. In contrast to the mixture obtained in **C1**, the success of **C3** implied that the longer

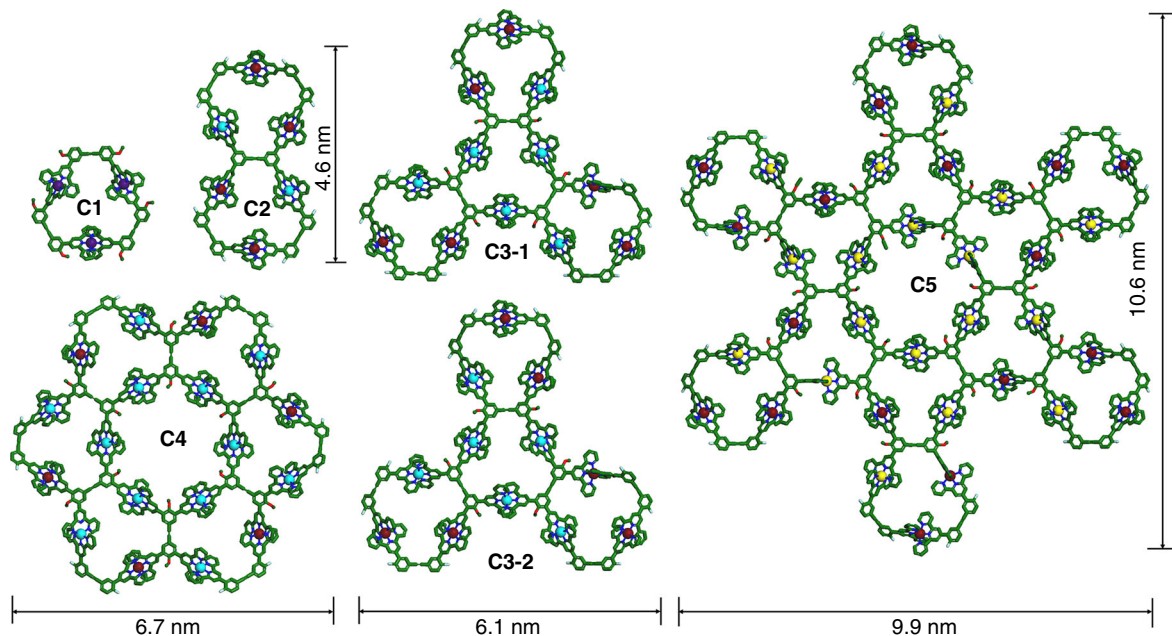

**Fig. 1** Energy-minimized structures of supramolecular cycles **C1**–**C5**. **C1**: violet = Fe; **C2**: cardinal = Ru, blue = Zn; **C3**–**1 & C3**–**2**: cardinal = Ru, blue = Zn; **C4**: cardinal = Ru, blue = Zn; **C5**: cardinal = Ru, yellow = Cd. Alkyl chains were omitted for clarity

**Fig. 2** Design of linear metal-organic ligands **L1**–**L5** and corresponding supramolecules **C1**–**C5**. **C1**: violet = Fe; **C2**: cardinal = Ru, blue = Zn; **C3**–**1 & C3**–**2**: cardinal = Ru, blue = Zn; **C4**: cardinal = Ru, blue = Zn; **C5**: cardinal = Ru, yellow = Cd

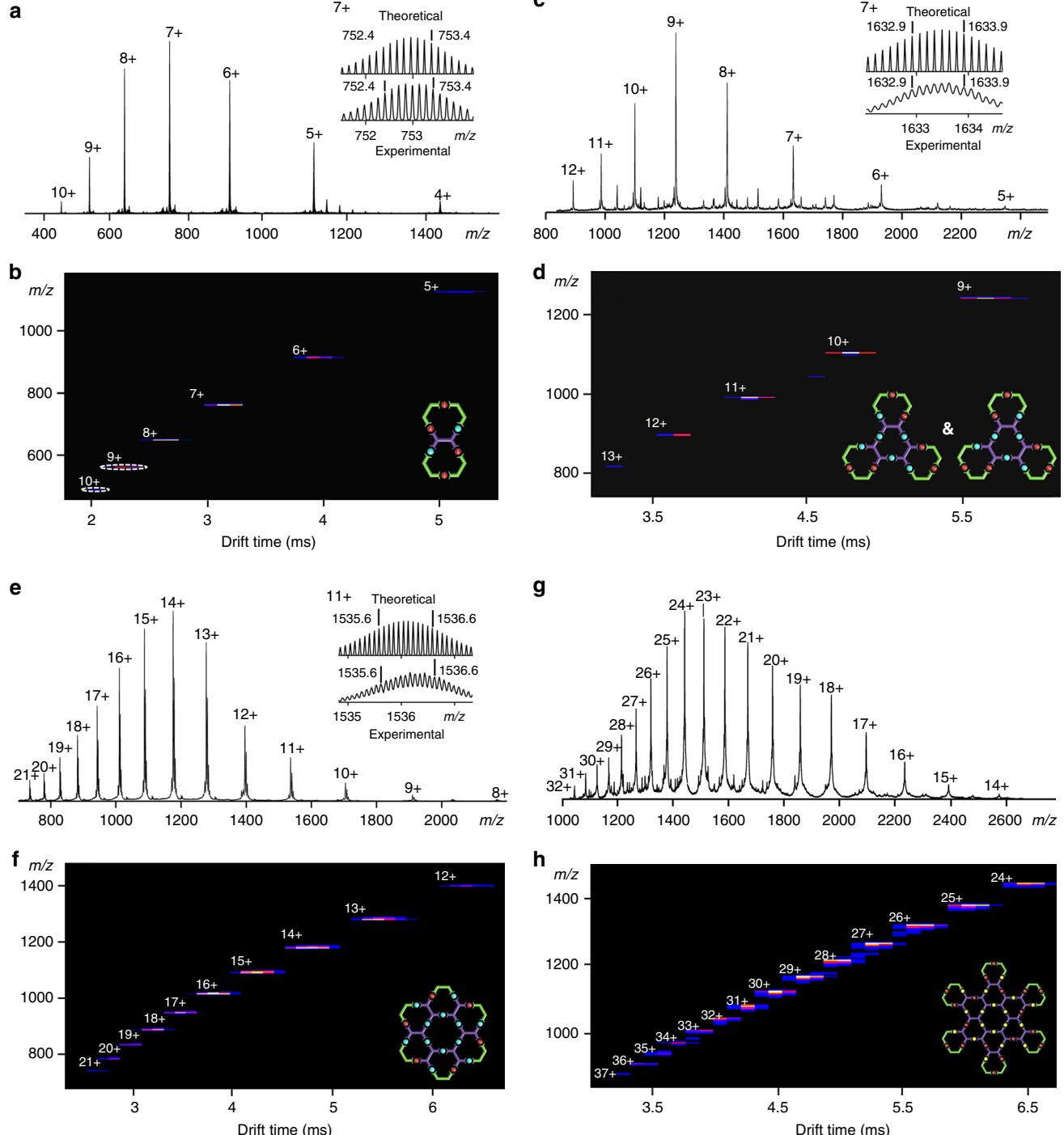

**Fig. 3** Mass spectrometry for supramolecules **C2**–**C5**. Electrospray ionization–mass spectrometry (ESI-MS) for **C2** (**a**); **C3** (**c**); **C4** (**e**) and **C5** (**g**). Traveling wave Ionmobility–mass spectrometry (TWIM-MS) for **C2** (**b**); **C3** (**d**); **C4** (**f**) and **C5** (**h**)

sequence increased specificity and rigidity of the ligand backbone, which led to the successful self-assembly. Interestingly, [1]H NMR and [19]F NMR spectra both showed significant signs of isomers, **C3–1** and **C3–2** (Supplementary Figures 76, 78). This is because the ligand can flip over with different orientation and form the self-assembly with the same shape, but different arrangements of Ru(II) and Zn(II) on the outer layer.

In order to get a single structure, another ligand **L3'** with three Ru(II) on the outer layer was synthesized. By introducing an extra Ru(II) ion into the system, which connected the start and end terminals of 'ABB' sequence, a symmetrical ligand **L3'** was obtained with higher rigidity, which ensured the exact geometry of the final self-assembly. Both ESI-MS and TWIM-MS showed the molecular information of desired structures, and [1]H NMR, 2D COSY & NOESY spectra further supported the formation of a single product **C3'** (Supplementary Figures 83–89).

Following the same design principle, introducing '**B**' on both sides would lead to a closed structure. Elongation of the sequence from 'ABB' to 'BBABB' gave a linear ligand **L2**. Similar to **L3**, the synthesis was initiated with coordination between '**A**' and two equivalents of '**B**', followed by Sonogashira coupling with two equivalents of Ru complex. During the first step of coordination,

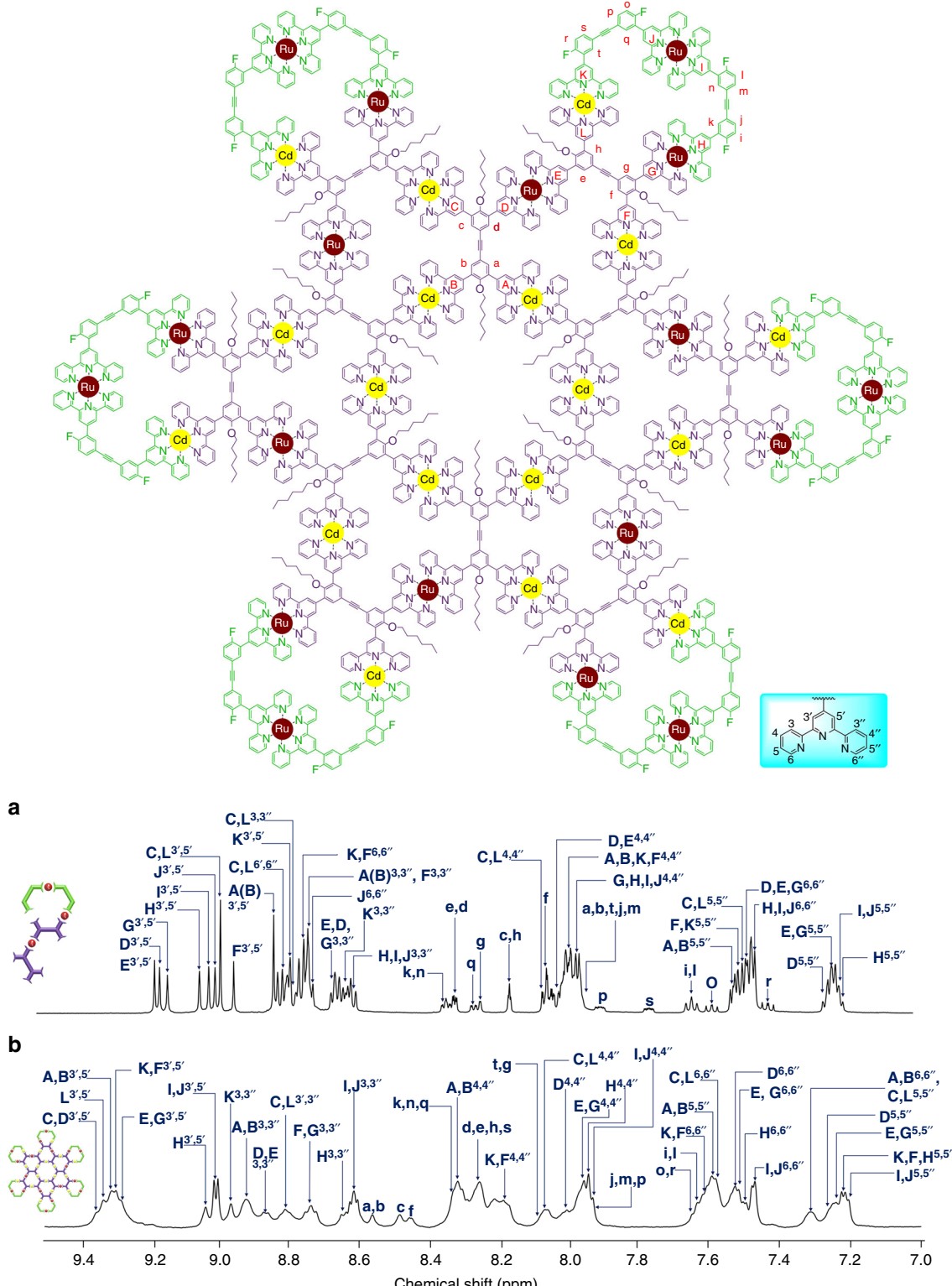

**Fig. 4** $^1$H NMR spectra (aromatic region). **a** L5 (600 MHz, CD$_3$CN, 300 K); **b** C5 (600 MHz, CD$_3$CN, 300 K)

both *cis* and *trans* isomers were formed (Supplementary Figure 4). The *trans* one as the major product was eventually isolated via column chromatography on aluminum oxide, while the *cis* product was unable to be isolated due to the negligible fraction (Supplementary Figure 104). The successful isolation of configurational isomers enabled the synthesis of L2 as 'BBABB' sequence without isomerization. Afterwards, mixing Zn(II) and

L2 in 1:2 ratio in CHCl$_3$/MeOH at 50 °C for 8 h, a set of peaks with clear isotope patterns was observed on ESI-MS, and was consistent with theoretical calculations. TWIM-MS clearly showed a set of peaks with continuous charges from 5+ to 10+, indicating the uniqueness of the assembly (Figs. 3a, b). According to $^1$H NMR, the 6, 6" proton on the metal-free terpyridines in ligand showed diagnostic upfield shifts after the

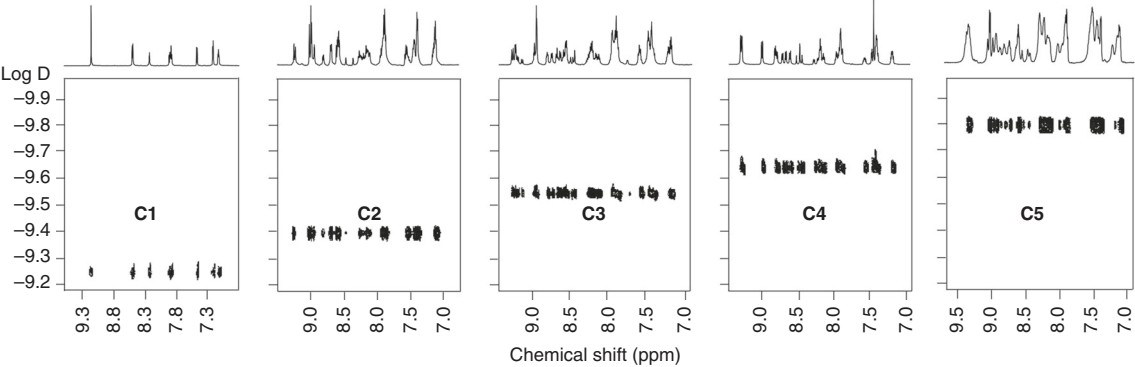

**Fig. 5** DOSY NMR spectra of supramolecules **C1**–**C5**

coordination with Zn(II) due to the electron shielding effect (Supplementary Figure 107). Only one set of peaks was observed at 3.70 ppm for **C2**, indicating the single conformation of self-assembly (Supplementary Figure 69). [19]F spectrum for the supramolecule showed three sets of peaks with 1:1:2 ratio (Supplementary Figure 71), suggesting the symmetry of the structure after coordination. If we were able to isolate precursor **15-cis** for further coupling reaction to obtain *cis* **L2**, the self-assembly of *cis* **L2** with Zn(II) would give similar C2-like structure (Supplementary Figure 112a).

Based on the success of previous generations, **C5** with giant size and high complexity was finally achieved. **L5** has a sequence of '**AABB**', which was synthesized via two steps of coordination and one step of Sonogashira coupling (Supplementary Figure 7). Similar to **L2**, isomeric compounds during the second step coordination was successfully isolated via column chromatography. Note that we attempted to synthesize **L5'** using a direct 'end-capping' strategy similar to **L3'** (Supplementary Figure 111). However, undesired products were generated along with **L5'** due to the flexibility and multiple coordination sites. We were unable to isolate **L5'** for further self-assembly. Using **L5** with linear structure to assemble, the self-assembly can be controlled by intramolecular and intermolecular complexation processes with Cd(II), which has high reversibility and low coordination strength to achieve energy favorable structure. The intramolecular complexation resulted in **C1**-like cycle, followed by intermolecular complexation to further form **C5**. This process is reminiscent of the folding and self-assembly of proteins. Similarly, if we were able to synthesize *cis* **L5**, the further self-assembly would construct **C5**–like architecture (Supplementary Figure 112b). [19]F NMR of the ligand showed three sets of peaks with an integration ratio of 2:1:1 (Supplementary Figure 58), which indicated the formation of a pure ligand **L5** without isomerization. In the structure of **C5**, '**A**' located in the center and formed a structure similar to **C4**, while '**B**' was left on the outside for end-capping. Considering the large molecular weight and increasing number of coordination sites, we chose Cd(II) as the self-assembly metal ion, which has higher reversibility and self-correction ability than Zn(II) for assembling of large and complex architectures[46]. After mixing **L5** and Cd(II) in 1:3 ratio at 50 °C for 8 h, mass spectrometry was first applied to characterize the assembly. ESI-MS displayed a series of peaks with continuous charges from 14+ to 32+, corresponding to molecular weight 38,066 Da, which is among one of the largest 2D metallo-supramolecules ever reported[11, 47, 48]. These signals were consistent with theoretical values of the mass-to-charge ratio of each charge state (Fig. 3g). Only one set of peaks on TWIM-MS excluded the formation of other isomers or conformers (Fig. 3h). Isotope patterns of each charge state, however, was not obtained after numerous attempts, possibly due to the high molecular

weight beyond resolution of our ESI-TOF mass spectrometer. [1]H NMR showed broad peaks, owning to the giant structure as well as similar chemical environments for some terpyridines after coordination. Despite that, we were still able to observe a well-resolved downfield shift of the free terpyridinyl-3', 5' and upfield shift of the free terpyridinyl-6,6" protons in the coordination with Cd(II) (Fig. 4). The peaks at δ 3.8 ppm had an integration ratio of 3:1, owing to the alkyl chains located in the different chemical environment of the inner and outer layer of the rings (Supplementary Figure 97). [19]F NMR showed only one broad peak at -115.31 ppm (Supplementary Figure 99), possibly due to the similar environment of the outer layer.

**Characterization of size based on DOSY-NMR, collision-cross section, TEM and AFM.** DOSY-NMR was applied to provide size information of the five structures. As shown in Fig. 5, **C1**–**C5** exhibit a single narrow diffusion band, respectively, indicating the discrete species were assembled. Diffusion coefficient decreased gradually from **C1** to **C5**, suggesting the increase in size. The experimental radius was also calculated using the modified Stocks–Einstein equation based on the oblate spheroid model (Supplementary Figures 113–118)[49–51]. Moreover, the experimental collision-cross section deduced from TWIM-MS[52] for the supramolecules were consistent with theoretical value calculated via MOBCAL[53] (Supplementary Table 1). TEM and AFM characterization also provided more evidence of the complexes regarding the size. Uniform dots were observed on TEM images for **C3**–**C5** with the measured size in accordance with theoretical modeling (Supplementary Figure 23). The AFM image also showed uniform dots for **C4** and **C5** with height around 0.75 nm, which is in good in agreement with the actual height of all the supramolecules (Supplementary Figure 24). The diameter for the single dots on AFM was inaccurate due to investable tip broadening effect[54].

**Hierarchical self-assembly of C4 and C5 in solution and on solid-liquid interface.** On the other hand, considering previous studies that the pre-assembled supramolecules may undergo hierarchical self-assembly based on either intermolecular or molecular-substrate interactions to form ordered nanostructures[55], we investigated the hierarchical self-assembly behavior of the supramolecules in hand. Since the rigid backbones contain multiple aromatic rings which possibly have π-π interactions with each other, the supramolecules are prone to stack with each other to form hierarchical self-assembled nanotubes[56]. We chose **C4** and **C5** as the representatives to verify our speculation. After a slow vapor diffusion of diethyl ether into the solution of **C4** and **C5** in DMF, regular stacked nanotubes were obtained and clearly observed under TEM (Figs. 6a–d). The

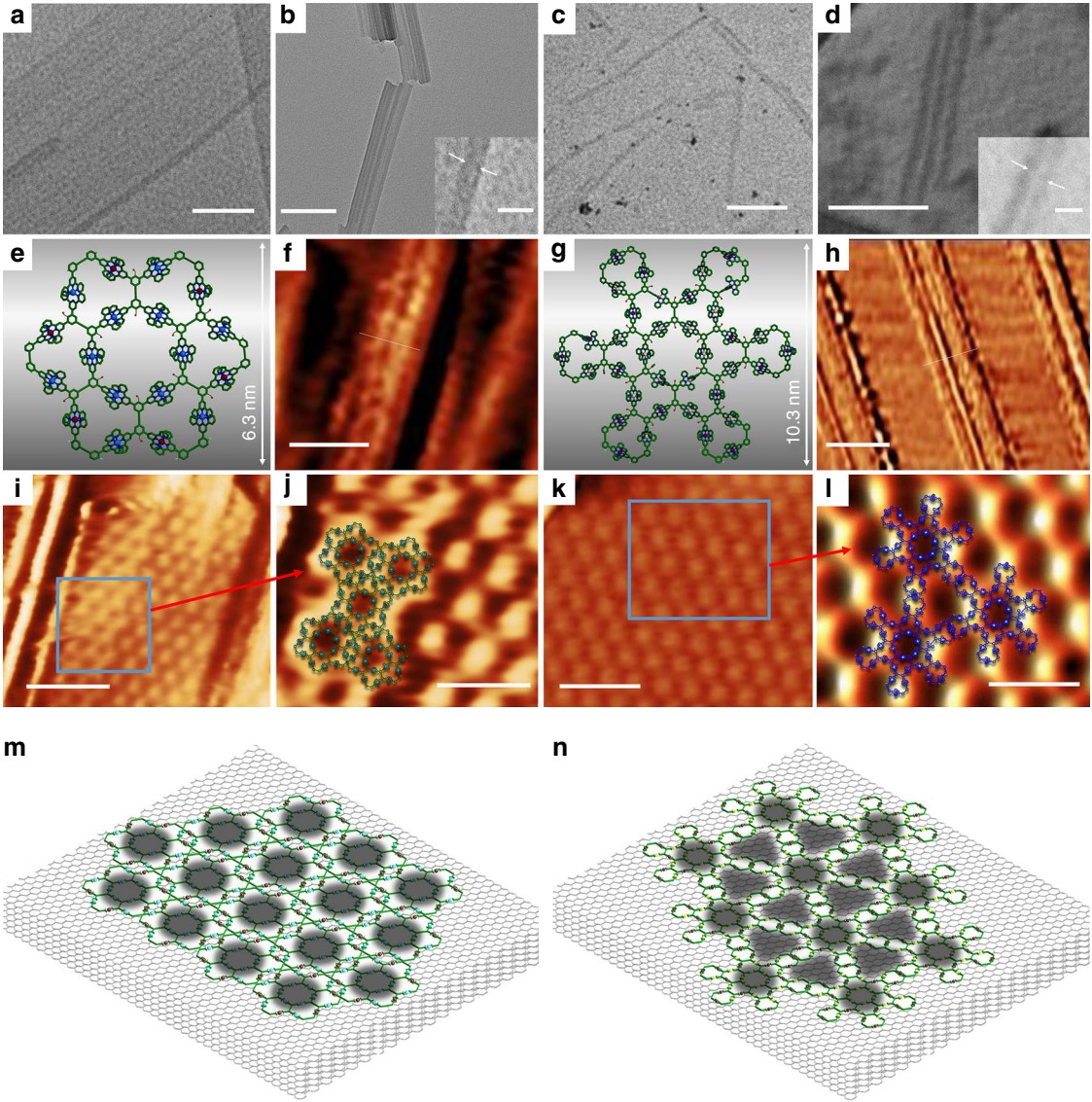

**Fig. 6** TEM & STM imaging of self-assembled nanotubes, nanoribbons, and nanosheets. TEM images of nanotubes assembled by **a** C4 (scale bar, 40 nm), **b** C4 (scale bar, 100 nm and 10 nm for zoom-in), **c** C5 (scale bar, 100 nm) and **d** C5 (scale bar, 100 nm and 10 nm for zoom-in); Energy-minimized structures for **e** C4 and **g** C5 (alkyl chains were omitted for clarity); STM images of self-assembled nanoribbons on HOPG surface of **f** C4 (scale bar, 10 nm) and **h** C5 (scale bar, 50 nm); 2D STM images of self-assembled nanosheets on HOPG surface of **i** C4 (scale bar, 25 nm) and **k** C5 (scale bar, 25 nm); 3D STM images from the highlighted area of **j** C4 (scale bar, 10 nm) and **l** C5 (scale bar, 10 nm); proposed patterns of **m** C4 and **n** C5 on HOPG surfaces (dark gray spots represent dark region observed on STM)

diameters of the tubes measured were comparable with single molecular width on theoretical modeling (Figs. 6e, g). Instead of forming single tubular nanostructures, the majority of the nanotubes observed were further stacked together, which possibly came from the self-assembly among the nanotubes with the ABAB type of packing[56].

Besides the formation of supramolecular nanotubes in solution, the behavior of supramolecules self-assemble on liquid–solid interfaces was even more attractive[57]. Previous researches suggested that this type of supramolecules with alkyl chains have at least two kinds of interactions with highly oriented pyrolytic graphite (HOPG) surface: (1) The π- π interactions between the aromatic rings and HOPG surface;[58] (2) The alkyl chains which could further induce self-assemblies on HOPG surfaces[59]. With these interactions, we might be able to observe self-assembly behavior of C4 and C5

on solid–liquid interfaces. After dropcasting of a solution of C4 on freshly cleaved HOPG surface and washed with water, single strand supramolecular metal-organic nanoribbons (SMONs) were obtained and further visualized under STM. The width of the SMONs was equal to the physical diameter of a single molecule C4 as simulated (Fig. 6f). Some of the SMONs bounded together and formed aggregates (Supplementary Figure 22). With longer deposition time of supramolecule solution on HOPG, we even observed the formation of supramolecular nanosheets through the neat arranged supramolecules (Figs. 6i, j). The outline of C4 was imaged as bright part, while the big hole in the center of the molecule was clearly observed (as shown in dark). The gaps between each supramolecule could not be clearly observed, perhaps because the supramolecules were very closely packed with each other (Fig. 6m).

The **C5** molecule showed similar behavior of forming SMONs with single molecular width (Fig. 6h). Besides, aggregated uniform SMONs with several molecular widths were also observed (Supplementary Figure 22). Similar to the nanosheets in **C4**, a large area was observed with **C5** well assembled on the HOPG surface, giving uniform molecular pattern (Figs. 6k, l). In contrast to **C4**, the gaps between individual **C5** were also clearly observed (Fig. 6l). Considering the geometry, the ordered nanostructure was proposed to be a 'head-by-head' style based on **C5** molecules (Fig. 6n).

## Discussion

In summary, using linear metal-organic ligands, we have successfully generated a series of 2D discrete polycyclic structures **C1**–**C5** with increasing complexity based on the coordination-driven self-assembly. The order of the residues enhanced the specificity of metal-organic building blocks, and thus in turn giving self-assemblies with different complexity as well as structural uniformity. In principle, the self-assembly can be divided into intramolecular and intermolecular complexation processes. **C1** was assembled by intermolecular complexation and **C2** was assembled by intramolecular complexation. In contrast, **C3**, **C4**, and **C5** were constructed by intramolecular complexation to form **C1**-like rings on the outside along with intermolecular complexation to form the rest rings in the center. With the improvement of synthesis and separation, design and utilizing linear metal-organic ligand with more types of residues and longer chain will open another avenue toward the construction of more complicated architectures. The pre-assembled supramolecules with precisely-controlled shapes and sizes may find more promising application as 2D materials based on their hierarchical self-assembly behaviors both in solution and on the liquid–solid surface.

## Data availability

The authors declare that all data supporting the findings of this study are available within the paper and its supplementary information files.

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

## Acknowledgements

This research was supported by the National Science Foundation (CHE-1506722 to X.L.; ECCS-1609788 to B.X.), National Institutes of Health (1R01GM128037 to X. L.), National Natural Science Foundation of China (21528201 to X-Q.H.), and the Program for Science & Technology Innovation Talents in Universities of Henan Province (17HASTIT004 to X-Q.H.).

## Author contributions

X.L. conceived and designed the experiments. B.S., Y.Y., S L. and X-Q H. completed the synthesis. S.K., K.Z., A.R. and B.X. performed STM imaging. J.G., X-H.L., B.S. and M.W. conducted NMR experiments. B.S. finished MS and TEM characterization. L.W. and H. W. performed AFM imaging. B.S., B.X., X-H.L., L.W., H.W., Y.L. and X.L. analysed the data and wrote the manuscript. All the authors discussed the results and commented on and proofread the manuscript.
