## [Peer Review File · Nature Communications]

Reviewers' comments:

Reviewer #1 (Remarks to the Author):

The authors report on the self-assembly of a series of metallomacrocycles using the metalloligands composed of ligands A and B. The diverse molecular geometry of the metalloligands controlled by the connection sequence of A and B gave rise to different self-assembled structures. The authors correlate this result with the sequence of nucleotide bases in DNA. However, the stepwise metalloligand approach has been widely used by Newkome and Wang to create various metallosupramolecules. The similar structures have been reported by these groups and the author himself. For example, ligand L1 and its self-assembly behavior have been reported in *Eur. J. Org. Chem.* 2008, 3328-3334, but the reference was not cited in the manuscript. N2 is structurally close to the bowtie molecule in *J. Am. Chem. Soc.* 2012, 134, 7672-7675. The bimetallic metallotriangulane (very similar to structure N3) has also been established by the same approach (*Eur. J. Org. Chem.* 2016, 5091-5095). Therefore, the novelty is questionable.

The following issues should be addressed for improving the structural characterization. 1) The authors intentionally introduced fluorine as a marker to identify the structures by ¹⁹F NMR, but it turns out that the sensitivity is not good enough to do so. For example, there should be four kinds of ¹⁹F chemical environments for N2, but only one peak was observed. 2) N3 has two isomeric structures, but in Figures 1 and 2 only one structure is presented, which is a misleading illustration. 3) The TEM images of the hierarchical assemblies of N4 and N5 in Figure S24 showed unclear nanotubes (nanoribbons?). Only based on the selected morphology, the tube-like structures were directly attributed to the stacked models without providing further experimental evidence such as electron diffraction. 4) The molecular models for the arrangement of N4 and N5 in the nanosheets observed by STM are very ambiguous. The dark and bright areas are not consistent with the proposed models. The "head-by-head" style shown in Figure 6l is different from that in Figure 6n where the most outer rings are overlapped with each other. The proposed structures are only partially supported by the current experimental data.

Reviewer #2 (Remarks to the Author):

The manuscript "Self-Assembly of Supramolecular Norias Using Sequence-Specific Ligand" presents an interesting study on the ligand-programmed assembly of complex supramolecular structures. By using what they term "programming of sequence-specific ligands" they are able to essentially construct supramolecular building blocks that can further be assembled (via some planning and likely some luck) into heterometallic supramolecular constructs. Although the underlying design principles behind this work have been utilized by chemists across many fields for many years, the specific demonstration of stepwise and judicious building block choice in supramolecular chemistry is novel enough. Overall the work is of very high quality and warrants publication in *Nature Communications*, however, there are a number of points the authors can address in order to clarify, clean-up, and streamline their manuscript. These are outlined here:

1. It almost seems like there is word missing from the title, or maybe it should just be "...Sequence-specific Ligands".
2. Generally, the manuscript reads very well. At points it is a bit too conversational. For example, "Encouraged by the success of N4, we proceeded on our journey toward N3. This type of language should probably be avoided.

3. The authors note that Fe is used as it has stronger binding with terpyridine than Zn, this is most likely the case, but an added reference for terpyridine binding constants would help.

4. The manuscript would be much stronger given additional explanation of ligand design parameters. I assume these were chosen both for ease of synthesis and the alkyne group is needed to prevent free-rotation? Figure 2 makes it seem like the pieces are locked into place. If not, I could imagine making an infinite number of extended, polymeric structures.

5. The authors present a fairly compelling suite of characterization methods. Have they tried others? The materials are soluble, are crystal structures attainable for any of them? It is understandable if not for N2-N5 in the manuscript. What about the ligands and/or metallated pieces?

6. The diffusion coefficient plots are very useful, the authors should attempt to add mention of approximate hydrodynamic radii via this method.

Sincerely,
Eric D. Bloch

Reviewer #3 (Remarks to the Author):

This manuscript describes the assembly of diverse monomers possessing multiple terpyridine ligands and their reaction of different metal(II) ions. This is a continuation of their recent paper *Nature Commun.* (2018, 9:567). In general, the manuscript is very difficult to read because of the diverse ligands A and B giving L2, L3, L4, L5, L6, and L7, which then give products N1 – N7; also e.g., L6 is 'ABABB' and L8 is 'AABABB'.

Their title uses the term "Norias", which equates to an old fashion water wheel, and is, of course, a 3-dimensional construct, since it lifts water; this leads one to initially mentally paint the wrong picture. Although their constructs do form assemblies by stacking, these are polymeric in nature - not 3D. Thus, this obscure word is inappropriate.

And in their introduction, the relationship of these structures to DNA- and protein-based assemblies is certainly stretching the point – BTW, everyone does this relationship to introduce this topic but again there is limited relationship but then to conclude with "If we can solve the challenges in synthesis and separation, design and utilizing sequence-specific ligand[s] with more types of residues and longer chain[s] will open a new route toward the construction of more complicated metal-ligand self-assemblies with comparable complexity as DNA and proteins with no doubt." What happened to H-bonding and chirality?

There are also statements that are simply false, such as ..."the sequence of ligand[s] has seldom been addressed as a factor in the design of metallosupramolecules." This is the major objective in most papers in this arena, Stang, Constable, Lehn, Chan, Wang, et cetera.

Too many grammatical errors to delineate; it needs to be polished and too similar to their recent paper, noted above.

Response Letter

Reviewer 1

1. *The similar structures have been reported by these groups and the author himself. For example, ligand L1 and its self-assembly behavior have been reported in Eur. J. Org. Chem. 2008, 3328-3334, but the reference was not cited in the manuscript. N2 is structurally close to the bowtie molecule in J. Am. Chem. Soc. 2012, 134, 7672-7675. The bimetallic metallotriangulane (very similar to structure N3) has also been established by the same approach (Eur. J. Org. Chem. 2016, 5091-5095). Therefore, the novelty is questionable.*

Response: We thank the reviewer for such detailed comment. This manuscript was originally submitted to *Nature Chemistry*, which has reference limits up to 50. Therefore, we were only able to cite representative references. In our revised manuscript, all the mentioned and other related references have now been cited. The goal of this paper was to introduce the concept of “sequence-specific ligand”, which could possibly provide guidance in the systematic synthesis of metallo-supramolecular structures with precise geometrical control. To prove the assumption, we herein synthesized five different supramolecules using ligands with different sequence. Despite that N1 to N3 (now named C1 to C3) were similar to the structures previously reported, all the five structures with increasing molecular weight and complexity could be well demonstrated via the same sequence-specific system. Supramolecule N5 (now named C5) with molecular weight of 38K Da was also successfully obtained and the next generations of structures in this system were also predicted. Moreover, we have also studied heretical self-assemblies of giant supramolecules N4 (C4) and N5 (C5), which could potentially provide applications to these metallo-supramolecules in the future. All together, the novelty of this manuscript is emphasizing the concept of “sequence-specific ligand” rather than individual structure.

2. *The authors intentionally introduced fluorine as a marker to identify the structures by ¹⁹F NMR, but it turns out that the sensitivity is not good enough to do so. For example, there should be four kinds of ¹⁹F chemical environments for N2, but only one peak was observed.*

Response: To get better sensitivity, we run ¹⁹F NMR for the supramolecules on a Bruker AVANCE III 400 MHz NMR spectrometer equipped with a 5mm PABBO BB/19F-1H/D Z-

GRD probe, which has better sensitivity for fluorine. It should be noted that the sensitivity of ^{19}F relies more on the probe and shimming rather than magnetic field. All the samples have been carefully manually shimmed. Supramolecule N2 (which is now named C2) showed three sets of peaks with 1:1:2 ratio (updated Supplementary Figure 70), which is consistent with the three sets of peaks in the ligand (Supplementary Figure 29). Supramolecule N4 (which is now named C4) showed two sets of peaks with 1:1 ratio (Supplementary Figure 91), which is consistent with the two kinds of fluorine atoms in the structure. The other structures did not show significant improvement.

3. N3 has two isomeric structures, but in Figures 1 and 2 only one structure is presented, which is a misleading illustration.

Response: We would like to thank the reviewer for the correction. Both Figure 1 and Figure 2 have been corrected, showing N3 (which is now named C3-1 and C3-2) as isomers.

4. The TEM images of the hierarchical assemblies of N4 and N5 in Figure S24 showed unclear nanotubes (nanoribbons?). Only based on the selected morphology, the tube-like structures were directly attributed to the stacked models without providing further experimental evidence such as electron diffraction.

Response: To get clear images of the nanotubes, we performed high resolution TEM for the nanotubes from both N4 and N5 (now named C4 and C5). The nanostructures for C5 were mainly observed as single nanotube, for which the diameter is consistent with theoretical modeling. The nanotubes for C4 were mainly observed in bundles, which were packed by several single nanotubes. And the diameter of each nanotube is consistent with the diameter of supramolecule C4 as modeled, (see revised Supplementary Figure 24). We have also performed electron diffraction (Figure R1), however, were unable to obtain satisfactory diffraction data for any conclusive result. Compared to a previous study by Aida and co-workers (*Science* 2014, 344, 499), growing large area of high-quality ordered nanotubes suitable for either electron or X-ray diffraction is a major problem we need to overcome in future studies, although we tried many solvent systems for the nanotube growth.

Figure R1. (a) TEM image and zoom-in of single-strand nanotube of C5. (b) TEM image and electron diffraction for nanotubes of C4.

5. *The molecular models for the arrangement of N4 and N5 in the nanosheets observed by STM are very ambiguous. The dark and bright areas are not consistent with the proposed models. The “head-by-head” style shown in Figure 6l is different from that in Figure 6n where the most outer rings are overlapped with each other. The proposed structures are only partially supported by the current experimental data.*

Response: We thank the reviewer for catching up the discrepancy between Figure 6n (the proposed molecular model) and Figure 6l (the actual STM image). The outmost rings should not overlap with each other in the “head-by-head” style of the 2D assembly pattern. Rather there is a space among each three of the nearest neighbor molecules, as clearly shown in the STM image. The proposed model is corrected (revised Figure 6n).

Reviewer 2

1. *It almost seems like there is word missing from the title, or maybe it should just be “...Sequence-*

Specific Ligands”.

Response: We thank the reviewer for the correction. The title has been changed to “Self-assembly of Polycyclic Supramolecules Using Sequence-specific Ligands”.

2. *Generally, the manuscript reads very well. At points it is a bit too conversational. For example, "Encouraged by the success of N4, we proceeded on our journey toward N3. This type of language should probably be avoided.*

Response: We greatly appreciate the reviewer for the recommendation. We have carefully revised the manuscript and made corrections on grammatic errors as well as conversational languages, *i.e.*, avoiding usage of 1st person in experimental part. The sentence mentioned above have also been corrected to “The journey proceeded to C3 with the construction of ‘ABB’ sequence.”

3. *The authors note that Fe is used as it has stronger binding with terpyridine than Zn, this is most likely the case, but an added reference for terpyridine binding constants would help.*

Response: We thank the reviewer for the reminding. We have added reference (*Chem. Eur. J.* 2012, 18, 11569) accordingly.

4. *The manuscript would be much stronger given additional explanation of ligand design parameters. I assume these were chosen both for ease of synthesis and the alkyne group is needed to prevent free-rotation? Figure 2 makes it seem like the pieces are locked into place. If not, I could imagine making an infinite number of extended, polymeric structures.*

Response: We have added some descriptions of the synthesis of ligands in the manuscript (Page 5). In fact, the use of alkyne was the concern of both structure rigidity as well as synthesis, *e.g.*, solubility, separation, etc. Meanwhile, alkyne serves a perfect 180-degree linker and does not introduce extra hydrogen atoms compared to phenyl group. Such design helps simplify ¹H NMR spectrum.

5. *The authors present a fairly compelling suite of characterization methods. Have they tried others? The materials are soluble, are crystal structures attainable for any of them? It is understandable if not for N2-N5 in the manuscript. What about the ligands and/or metallated pieces?*

Response: We have tried growing single crystals for supramolecules. However, all slow vapor-diffusions gave either gel or precipitate, which were mainly observed as nanotubes or nanoribbons under TEM. Up to now, all the attempts for single crystal growth for ligands failed to give high quality crystals suitable for X-ray diffraction, possibly due to the multiple alkyl chains that reduced rigidity of supramolecules and affected the packing of the supramolecules. X-ray diffraction is definitely the best way to unambiguously determine molecular structure, which is what we need to focus on in future studies.

6. The diffusion coefficient plots are very useful, the authors should attempt to add mention of approximate hydrodynamic radii via this method.

Response: Following the success of previous studies (*Angew. Chem. Int. Ed.* 2008, 47, 2235; *Org. Biomol. Chem.* 2014, 12, 7932; *J. Am. Chem. Soc.* 2017, 139, 8174), we herein used the modified Stocks-Einstein equation to calculate the experimental radius of the supramolecule based on oblate spheroid model (Supplementary Figure 95).

Reviewer 3

1. Their title uses the term "Norias", which equates to an old fashion water wheel, and is, of course, a 3-dimensional construct, since it lifts water; this leads one to initially mentally paint the wrong picture. Although their constructs do form assemblies by stacking, these are polymeric in nature - not 3D. Thus, this obscure word is inappropriate.

Response: We would like to thank the reviewer for the comment. We have changed the word "Norias". The title has been changed to "Self-assembly of Polycyclic Supramolecules Using Sequence-specific Ligands".

2. In their introduction, the relationship of these structures to DNA- and protein-based assemblies is certainly stretching the point – BTW, everyone does this relationship to introduce this topic but again there is limited relationship but then to conclude with "If we can solve the challenges in synthesis and separation, design and utilizing sequence-specific ligand[s] with

more types of residues and longer chain[s] will open a new route toward the construction of more complicated metal-ligand self-assemblies with comparable complexity as DNA and proteins with no doubt." What happened to H-bonding and chirality?

Response: We thank the reviewer for the correction. In this paper, we introduce the concept of “sequence-specific ligands”, which was inspired by the sequence of DNA and proteins. However, DNA and protein are complex systems with multilevel structures involving multiple non-covalent interactions. Currently, artificial metallo-supramolecular systems were unable to reach such comparable complexity. The statement has been revised to: “With the improvement of synthesis and separation, designing and utilizing sequence-specific ligand with more types of residues and longer chain will open another route toward the construction of more complicated metal-ligand self-assemblies.”

3. There are also statements that are simply false, such as ..."the sequence of ligand[s] has seldom been addressed as a factor in the design of metallosupramolecules." This is the major objective in most papers in this arena, Stang, Constable, Lehn, Chan, Wang, et cetera.

Response: We thank the reviewer for the correction. Such kind of problematic statement has been removed from the manuscript.

4. Too many grammatical errors to delineate; it needs to be polished and too similar to their recent paper, noted above.

Response: We thank the reviewer for the suggestion. The whole paper has been carefully proof read and the language have been polished.

Reviewers' comments:

Reviewer #1 (Remarks to the Author):

In the revised version, the authors did some corrections and added the re-measured ^{19}F NMR spectra for C2 and C4 as well as the calculation of hydrodynamic radii. However, the concept of "sequence-specific ligands" the authors emphasized is not sufficiently supported by the molecular design and assembled structures. For the ligand design, the strategy described in the manuscript is a commonly used metalloligand approach based on kinetically inert bispypy-Ru(II) complexes. Numerous examples can be found in literature. In principle, the self-assembly can be divided into intramolecular and intermolecular complexation reactions. The intramolecular complexation resulted in C1-like cycles, which could be found in C2, C3, and C5. In the case of C3, L3 helped constrain the conformation but still gave a mixture of isomers. The redundant/decorating trinuclear cycle containing two Ru(II) and one Cd(II) or Zn(II) could be generated intramolecularly and propagated to a fused ring like C2 by increasing the number of intramolecular complexation positions. The decorating part can be elongated liberally as the proposed assemblies C6 and C7. Notably, C6 will have the same isomeric problem as that in C3. On the other hand, the intermolecular complexation is responsible for creating core structures (C1 and C4) in this system, which is more meaningful in terms of exploring new self-assembly processes, but the structure of C1 has been reported, and C4 is very similar to the previously reported hexagon wreath and assembled by the same coordination principle. The metalloligands with different geometries composed of two subunit types apparently should be built by varying the connection sequence, so the description of "sequence-specific ligands" is very tricky. In addition, the subunit orientation should also play a role in the self-assembly, which is not discussed. For example, will the constitutional isomers (cis isomers) of L2 and L5 give rise to the same polycyclic structures? The intra- and intermolecular complexation may be disrupted due to the varied subunit arrangement. Again, the nanostructures generated from C4 and C5 cannot be directly correlated to the stacking models in Figures 6a and 6c without giving persuasive experimental evidence or computational support.

Reviewer #2 (Remarks to the Author):

The authors largely addressed previous concerns I had with the manuscript. With these revisions in mind I believe the manuscript is suitable for publication in Nature Communications.

Response Letter to Reviewer 1's Comments

1. The concept of “sequence-specific ligands” the authors emphasized is not sufficiently supported by the molecular design and assembled structures. For the ligand design, the strategy described in the manuscript is a commonly used metalloligand approach based on kinetically inert bipyridine-Ru(II) complexes. Numerous examples can be found in literature. In principle, the self-assembly can be divided into intramolecular and intermolecular complexation reactions. The intramolecular complexation resulted in C1-like cycles, which could be found in C2, C3, and C5. In the case of C3, L3 helped constrain the conformation but still gave a mixture of isomers. The redundant/decorating trinuclear cycle containing two Ru(II) and one Cd(II) or Zn(II) could be generated intramolecularly and propagated to a fused ring like C2 by increasing the number of intramolecular complexation positions. The decorating part can be elongated liberally as the proposed assemblies C6 and C7.

Response: We would like to thank the reviewer for such detailed suggestion, which inspired us to a more in-depth thinking of our work. We have added discussion about inter- and intramolecular complexation in the manuscript. We also changed the title without emphasizing “sequence specific ligand” as a new design strategy.

1. End-capping approach based on the coordination with Ru(III) complex followed by reduction

Newkome GR, et. al. *Angew. Chem. Int. Ed.*, **1999**, 38, 3717;
Newkome GR, Wang P, et. al. *Science*. **2006**, 312, 1782;
Newkome GR, Li X, et. al. *Chem. Eur. J.* **2012**, 18, 11569;
Newkome GR, Li X, et. al. *Eur. J. Org. Chem.* **2013**, 2013, 3640;
Newkome GR, Wang P, et. al. *Chem. Commun.* **2015**, 51, 5766;
Newkome GR, et. al. *Eur. J. Org. Chem.* **2016**, 2016, 5091;
Newkome GR, et. al. *J. Am. Chem. Soc.* **2017**, 139, 15652;
Newkome GR, et. al. *Dalton Trans.* **2018**, 47, 7528.

2. Suzuki coupling reaction on terpyridine-Ru(II) complex

Wang P, Li X, et. al. *J. Am. Chem. Soc.*, **2016**, 138, 10041;
Wang P, Li X, Newkome GR, et. al. *Chem. Commun.*, **2016**, 52, 9773;
Wang P, Li X, Newkome GR, et. al. *Nat. Commun.*, **2017**, 8, 15476;
Wang P, Li X, Newkome GR, et. al. *Angew. Chem. Int. Ed.*, **2017**, 56, 11450;
Wang P, Li X, et. al. *Chem. Commun.*, **2017**, 53, 6732;
Li X, et. al. *J. Am. Chem. Soc.* **2017**, 139, 8174.

3. Sonogashira coupling reaction on terpyridine-Ru(II) complex (This work)

Figure R1. Three approaches to terpyridine-Ru(II) metal-organic ligand.

Indeed, using terpyridine-Ru(II) metal-organic ligands as building blocks is commonly used in terpyridine-based self-assembly. Here we categorize the preparation of terpyridine-Ru(II) metal-organic ligands into three approaches as shown in Figure R1, including 1) end-capping approach based on the coordination with Ru(III) complex followed by reduction; 2) Suzuki coupling reaction on terpyridine-Ru(II) complex; 3) Sonogashira coupling reaction on terpyridine-Ru(II) complex. End-capping approach was first developed to prepare symmetric metal-organic ligand. With the introduction of Suzuki coupling reaction, short linear asymmetric metal-organic ligand could be obtained for single point or multiple points with the same $-B(OH)_2$ reactant. The synthesis of long sequences needs asymmetric step-wise coupling reactions on terpyridine-Ru(II) complexes. Without appropriate protection and deprotection groups, it is very challenging to use Suzuki coupling reaction for longer asymmetric metal-organic ligand synthesis.

With the help of $-TMS$ protection and deprotection of alkyne, we successfully synthesized sequences “**AB**, **ABB**, **AABB**, and **BBABB**” using Sonogashira coupling reaction. This synthetic approach is similar to the protection and deprotection in peptide synthesis. With such a powerful method, we should be able to synthesize even longer metal-organic ligand for the self-assembly of giant metallo-supramolecules with increasing complexity. Therefore, the strategy used in our study is different from previous study.

Note that we attempted to synthesize **L5'** using a direct ‘end-capping’ strategy similar to **L3'** (Figure R2, Supplementary Figure 110). However, undesired products were generated along with **L5'** due to the flexibility and multiple coordination sites. We were unable to isolate desired **L5'** for further self-assembly. Using **L5** with linear structure to assemble, the self-assembly can be controlled by intramolecular and intermolecular complexation processes with Cd(II), which has high reversibility and low coordination strength to achieve energy favorable structure. The intramolecular complexation resulted in C1-like cycle, followed by intermolecular complexation to further form **C5**. This process is reminiscent of the folding and self-assembly of proteins.

Figure R2. Three approaches to terpyridine-Ru(II) metal-organic ligand.

2. On the other hand, the intermolecular complexation is responsible for creating core structures (**C1** and **C4**) in this system, which is more meaningful in terms of exploring new self-assembly processes, but the structure of **C1** has been reported, and **C4** is very similar to the previously reported hexagon wreath and assembled by the same coordination principle.

Response: Newkome and coworkers (*Eur. J. Org. Chem.* 2008, 3328) reported four macrocycles through direct self-assembly of bisterpyridine ligands with Fe(II) as shown in Figure R3. Our **C1** structure is one of them. However, the goal of our study in this manuscript is not repeating Newkome's study with slight modification. Instead, we use linear metal-organic ligands to continuously increase the complexity of assemblies. For instance, our **C5** supramolecule contain 12 **C1**-like structure with a central hexagon. In the early stage, supramolecular chemists focused on the self-assembly methodology development through design and assembly of polygons and polyhedrons. Up to date, the design of metallo-supramolecules has matured beyond the proof of principles and is ready to face more challenges with respect to the complexity of assembled architectures rather than relatively simple polygons and polyhedrons. Therefore, we demonstrate that we can use basic polygon as repeating unit to create more complex architecture. **C1** is just a model system rather than our major focus in this study.

Figure R3. Newkome's previous study vs. **C1** and **C5**. Adapted from *Eur. J. Org. Chem.* 2008, 3328-3334. with permission. Copyright 2008 Wiley-VCH.

We also compared our hexagon wreath (*J. Am. Chem. Soc.*, 2014, 136, 6664) with **C4** as shown in Figure R4. Obviously, hexagon wreath was assembled with organic tetratopic terpyridine ligand with Zn(II) directly; **C4** was assembled using metal-organic ligand **AB** through both intermolecular and intramolecular complexations with Zn(II). The complexity of **C4** is higher than hexagon wreath. More importantly, we were unable to assemble **C4** with direct self-assembly of **A**, **B** and Zn(II) as we did for hexagon wreath.

Figure R4. Hexagon wreath supramolecule vs. **C4** in our study. Adapted from *J. Am. Chem. Soc.* **136**, 6664-6671 with permission. Copyright 2014 American Chemical Society.

3. The decorating part can be elongated liberally as the proposed assemblies **C6** and **C7**..... Notably, **C6** will have the same isomeric problem as that in **C3**.

Response: Besides **L4**, any sequence starting with ‘**AB**’ would give isomers due to the symmetric nature of the **C1**-like ring. In one of our ongoing researches, we especially focus on the study of isomers at single-molecule level using high resolution microscopy. In order to make this manuscript more concrete, we removed **C6** and **C7** from this manuscript to focus on **C1-C5**.

4. In addition, the subunit orientation should also play a role in the self-assembly, which is not discussed. For example, will the constitutional isomers (*cis* isomers) of **L2** and **L5** give rise to the same polycyclic structures? The intra- and intermolecular complexation may be disrupted due to the varied subunit arrangement.

Response: We added the discussion of subunit orientation for the self-assembly of **C3-1** and **C3-2**. Due to the small fraction, the *cis*- isomers of building blocks were unable to be isolated via column. Considering the rigidity of backbones and reversibility of metal ions during self-assembly, we speculate that the *cis*- isomers would also give the desired polycyclic structures (Figure R5). The corresponding discussion was added to the manuscript.

Figure R5. Self-assembly of *cis*- a) **L2** and b) **L5** for **C2**-like and **C5**-like structures, respectively.

5. Again, the nanostructures generated from C4 and C5 cannot be directly correlated to the stacking models in Figures 6a and 6c without giving persuasive experimental evidence or computational support.

Response: We thank the reviewer for the comment. We have removed the model and replaced with two extra TEM images of nanotubes, see revised Figure 6a and 6c.

REVIEWERS' COMMENTS:

Reviewer #1 (Remarks to the Author):

In the revised manuscript, the authors changed the title to "Self-Assembly of Polycyclic Supramolecules Using Linear Metal-Organic Ligands" to attenuate the concept of sequence-specific ligand design. Instead of underlining the original ligand design, the protection-deprotection approach to the preparation of tpy-Ru(II) metal-organic ligands is emphasized. Since the whole story is rewritten, as discussed in the point-by-point response, Figure R1 should be included in the manuscript to address this point.

In addition, the molecular models for the nanostructures observed in the TEM images have been removed, so the sentence of "The inner diameter of the tubes measured were consistent with single molecular width on theoretical modeling, suggesting that the nanotubes were possibly formed through a layer-by-layer stacking" (line 216) should be also removed for consistency.

In the caption of Figure 2, "ligands L1-L7" should be changed to "ligands L1-L5". The molecular structures reported here are very complicated, particularly for ligand L5 and supramolecule C5. The structural characterization and spectroscopic assignments should be done in a more careful way. There are numerous mistakes in the characterization part. For example, the assignments in Supplementary Figures 9 and 12 are incorrect.

Response Letter to Reviewer 1's Comments

1. In the revised manuscript, the authors changed the title to “Self-Assembly of Polycyclic Supramolecules Using Linear Metal-Organic Ligands” to attenuate the concept of sequence-specific ligand design. Instead of underlining the original ligand design, the protection-deprotection approach to the preparation of tpy-Ru(II) metal-organic ligands is emphasized. Since the whole story is rewritten, as discussed in the point-by-point response, Figure R1 should be included in the manuscript to address this point.

Response: We would like to thank reviewer 1 for the suggestion. Consideration the large figure size we herein put Figure R1 in supplementary material as “Supplementary Figure 1”. The references were cited in supplementary references. We have also added corresponding discussion in the manuscript on Page 4.

2. In addition, the molecular models for the nanostructures observed in the TEM images have been removed, so the sentence of “The inner diameters of the tubes measured were consistent with single molecular width on theoretical modeling, suggesting that the nanotubes were possibly formed through a layer-by-layer stacking” (line 216) should be also removed for consistency.

Response: We would like to thank reviewer 1 for the comment. The sentence has been modified to “The diameter of the tubes measured were comparable with single molecular width on theoretical modeling”.

3. In the caption of Figure 2, “ligands L1-L7” should be changed to “ligands L1-L5”.

Response: We thank reviewer 1 for the comment. The caption has been corrected to “Ligands L1-L5”.

4. The molecular structures reported here are very complicated, particularly for ligand L5 and supramolecule C5. The structural characterization and spectroscopic assignments should be done in a more careful way. There are numerous mistakes in the characterization part. For example, the assignments in Supplementary Figures 9 and 12 are incorrect.

Response: We would like to thank reviewer 1 for catching up these mistakes. The mislabeled charge states in Supplementary Figures 9 and 12 (Now Supplementary Figures 10 and 13) have now been corrected. We have carefully checked all the NMR assignments. Some peaks with split were signed individually instead of gathering all the labels together to our best extent, i.e., tpy- $H^{4,4}$, tpy- $H^{5,5}$, tpy- $H^{6,6}$ for L2, L5, C2, C4 and C5. Also, some assignments were corrected after careful check with NOESY and COSY spectra. The other characterization data have also been double checked carefully.